# Targeting Underlying Inflammation in Carcinoma Is Essential for the Resolution of Depressiveness

**DOI:** 10.3390/cells12050710

**Published:** 2023-02-23

**Authors:** Milica M. Borovcanin, Katarina Vesić, Dragana Arsenijević, Maja Milojević-Rakić, Nataša R. Mijailović, Ivan P. Jovanovic

**Affiliations:** 1Department of Psychiatry, Faculty of Medical Sciences, University of Kragujevac, 34000 Kragujevac, Serbia; 2Department of Neurology, Faculty of Medical Sciences, University of Kragujevac, 34000 Kragujevac, Serbia; 3Center for Molecular Medicine and Stem Cell Research, Faculty of Medical Sciences, University of Kragujevac, 34000 Kragujevac, Serbia; 4Faculty of Physical Chemistry, University of Belgrade, 11000 Belgrade, Serbia; 5Department of Pharmacy, Faculty of Medical Sciences, University of Kragujevac, 34000 Kragujevac, Serbia

**Keywords:** carcinoma, depression, acute inflammation, chronic inflammation, drug-treatment

## Abstract

In modern clinical practice and research on behavioral changes in patients with oncological problems, there are several one-sided approaches to these problems. Strategies for early detection of behavioral changes are considered, but they must take into account the specifics of the localization and phase in the course and treatment of somatic oncological disease. Behavioral changes, in particular, may correlate with systemic proinflammatory changes. In the up-to-date literature, there are a lot of useful pointers on the relationship between carcinoma and inflammation and between depression and inflammation. This review is intended to provide an overview of these similar underlying inflammatory disturbances in both oncological disease and depression. The specificities of acute and chronic inflammation are considered as a basis for causal current and future therapies. Modern therapeutic oncology protocols may also cause transient behavioral changes, so assessment of the quality, quantity, and duration of behavioral symptoms is necessary to prescribe adequate therapy. Conversely, antidepressant properties could be used to ameliorate inflammation. We will attempt to provide some impetus and present some unconventional potential treatment targets related to inflammation. It is certain that only an integrative oncology approach is justifiable in modern patient treatment.

## 1. Introduction

Treating patients with cancer is challenging. In modern clinical practice and research on behavioral changes in patients with oncologic problems, there are several one-sided approaches to this problem. Oncologists are concerned in great detail with localization of the primary oncologic process, pre- and post-operative care, protocols for chemotherapy and radiation therapy, and monitoring for recurrence. However, it is quite common for psychiatrists to be involved in some of the phases of integrative treatment. Mental predisposition could be discussed in the etiology of various carcinomas [1]. Mental disturbances could be a consequence of the patient’s awareness of the illness onset and its possible impact on the patient’s overall quality of life, or they may follow somatic perturbations and be an impact of the various therapies applied [2,3]. Mental disorders could also induce cancer recurrence [4].

In the new therapeutic strategies for the treatment of neuropathic pain as an oncological complication, it was very interesting to draw a parallel between the changes in the acute and chronic phases of pain and mental disorders management [5]. Inflammatory processes, both acute and chronic, are a hallmark of both oncological and mental disorders [6,7]. The exacerbation of somatic disorders and mental illnesses could reflect acute inflammation, whereas prolonged processes are related to chronic inflammation [8,9,10]. The question is if these landmarks could induce depressive symptomatology, and if so, to what extent and whether these phenomena should be treated as simple non-comorbid depression. In the up-to-date literature, there are a lot of useful pointers on the relationship between carcinoma and inflammation and between depression and inflammation. This review article aimed to compare and integrate these complex interactions in the same context of carcinoma and depression comorbidity. Further, we will try to use this information to potentially improve the clinical approach and discuss the importance of the resolution of inflammation as a new treatment strategy in the cooccurrence of carcinoma and depression.

## 2. Acute and Chronic Inflammation in Carcinoma

Inflammation represents the systemic host response to tissue damage. It is usually caused by injury, ischemia, infection, or chemical exposure [11,12]. Additionally, inflammation plays an important role in tissue repair, regeneration, and remodeling [13]. The inflammatory response involves the recruitment and action of the immune response [14].

Inflammation occurs in two stages, acute and chronic inflammation [13]. Acute inflammation is a part of innate immunity initiated by immune cells and lasts for a short time [15]. It serves as a defense against infection, tissue damage, and allergens. Receptors of innate immunity recognize the structures of microorganisms (pathogen-associated molecular patterns—PAMPs), but also molecules that are released from damaged host cells [16,17]. These molecules are called danger-associated molecular patterns (DAMPs) and represent proteins or nucleic acids that are not normally found outside the cell. The most important DAMPs include chromatin-associated protein high-mobility group box 1 (HMGB1), adenosine triphosphate (ATP), uric acid (UA), deoxyribonucleic acid (DNA), and degraded extracellular matrix (ECM)-like heparan sulfate and hyaluronan. PAMPs and DAMPs are recognized via pattern recognition receptors (PRRs) [16,17]. The term “alarmin” is today used as a synonym for DAMP [18].

In acute inflammation, pro-inflammatory mediators such as acute-phase proteins, prostaglandins, leukotrienes, oxygen- and nitrogen-derived free radicals, chemokines, growth factors, and cytokines that are released by immune defense cells locally at the site of inflammation cause neutrophil infiltration [19,20,21,22]. C-reactive protein (CRP), fibrinogen, and procalcitonin (PCT) are part of an innate immune response that is detectable in serum within a few hours of the initiation of inflammation [19]. They facilitate the inflammatory process and represent hallmarks of acute inflammation [19]. Subsequently, other cells of innate and adaptive immunity (e.g., macrophages and lymphocytes) are recruited to the inflammatory environment [14]. In response to DAMPs, innate immune cells secrete cytokines that mediate normal cellular processes and communication between leukocytes and other cells, but also regulate the host’s response to damage [21,22].

Cytokines can exert proinflammatory and anti-inflammatory effects both locally and systemically [23]. Activated cells of innate immunity produce the most important proinflammatory cytokines: interleukin (IL)-1, tumor necrosis factor-alpha (TNF-α), IL-6, IL-12, and IL-23 [23]. Conversely, the cells of adaptive immunity—activated T lymphocytes—produce interferon-gamma (IFN-γ) and IL-17 [21,22]. Some cytokines, such as IL-1α and IL-33, act as alarmins [18]. They are released from host cells as a result of injury or death and subsequently mobilize and activate immune cells [18].

The resolution of acute inflammation begins when PAMPs and DAMPs are no longer present [14]. However, if the pathogen cannot be completely eradicated or there is a constant source of self-antigens or a growing tumor that continuously disrupts tissue structure and induces the production of inflammatory cytokines, the second stage of inflammation, chronic inflammation, occurs [15]. Long-lasting chronic inflammation can lead to many chronic diseases including cardiovascular, respiratory, neurodegenerative diseases, and cancer via dysregulation of various signaling pathways [13,14,24].

When Rudolf Virchow described leukocytes within primary tumor tissue, a possible link between inflammation and cancer was established in the 19th century [25,26]. Today, it is obvious that inflammation plays an important role in the biology of tumors. Inflammation may play an anti- or pro-tumorigenic role. Acute inflammation in neoplastic tissues is indicative of an anti-tumor immune response [25,26]. In chronic inflammation, the inflammatory microenvironment facilitates cell mutations and proliferation leading to tumor development [27]. Alteration of several signaling pathways may contribute to the development of genetic and epigenetic changes in local tissue cells [28,29]. Additionally, chronic inflammation attenuates anti-tumor immunity and affects cell proliferation, death, senescence, DNA mutation, and angiogenesis [12,25,30,31,32]. The question remains whether the inflammation is a consequence of the anti-tumor immune response or whether the tumor arose in the setting of chronic inflammation.

## 3. Acute and Chronic Inflammation in Depression

Pro-inflammatory peripheral biomarkers elevation, a higher risk of depression in inflammatory and autoimmune diseases, the ability of immune mediators to induce depressive symptoms, and the fact that activated microglial cells reduce levels of serotonin and generate oxidative stress (OS) molecules all point to immune system involvement in the pathogenesis of depression [33]. Blood-brain barrier (BBB) permeability, the brain-gut axis, and the brain-fat axis bring systemic, particularly inflammatory, changes into the spotlight, not just central nervous system (CNS) disturbances in depression [34].

Specific depressive symptomatology was explored in correlation with inflammatory changes in the periphery. Majd et al. (2020) conducted a narrative review and indicated that there is an association between neurovegetative symptoms of depression, such as sleep problems, fatigue or loss of energy, appetite changes, and inflammation [35]. Increased inflammatory markers were measured in patients with major depressive disorder: IL-1β, IL-6, TNF-α, and CRP. Peripheral inflammation could signal the brain by leaky regions in the BBB, the cytokine transport system, and the vagus nerve. They based their conclusions on Capuron et al. (2002) [36], who demonstrated that IFN administration causes neurovegetative symptoms in the first two weeks, which are less responsive to antidepressant therapy, and depressed mood and cognitive symptoms later, which are responsive to antidepressants. Among other prominent theories of depression, the cytokine theory has played an important role in clinical practice [37]. Cytokines and peripheral immune cell counts could serve as biomarkers for distinct subgroups of inflamed depression and direct further treatment [38].

As recently noted in coronavirus disease (COVID-19), acute inflammation could be followed by behavioral changes termed “sickness behavior”, the resolution of which follows eradication of the infection, although in some cases psychotropic medications are required to resolve mental symptoms, particularly agitation [39]. It seems that some individuals have a predisposition to an exaggerated immune response to an infectious agent that could be harmful, not protective, and also lead to a later onset of depression [40]. The peripheral immune response is particularly exacerbated in depressive patients that are resistant to antidepressants [41]. Resilient animals do not display exacerbated immune responses following acute and chronic stress, suggesting that positive affectivity could buffer the negative impact of stress on immunity [42].

This hypersensitivity could be linked to the role of IL-6 as an important marker. A recently published first meta-analysis with a robust sample reported an adjusted association between IL-6 and future depression [43]. In addition, a small prospective association between depression and IL-6 was observed in both directions. If inflammation is prolonged and chronic, it is important to consider whether symptoms meet the threshold for a diagnosis of a depressive episode and require treatment. However, the elevation of IL-6 may be associated not only with chronic inflammation but also with other pathological processes that may also be observed in depression [43,44].

## 4. Animal Models of Inflammatory-Induced Depression in Carcinoma

The estimated high prevalence of depression in cancer patients and the insufficient data on the mechanisms by which tumors per se may alter brain functions, including mood and cognition, have engaged the preclinical research community to search for novel cancer-induced models. The main advantage of using animal models in research is the control of confounding variables that are difficult to control in the clinical setting and the ability to unravel mechanistic interactions between neural, immune, and inflammatory processes through which tumors alter brain function. Animal models provide a better explanation for the independent impact of tumor-associated biological processes on affective and cognitive symptoms, independent of cancer-associated stress and treatments.

Significant behavioral changes were found in mice with implanted tumors, characterized primarily by an increase in avoidance behavior and a decrease in immobility, defensive-submissive behavior, and non-social exploration [45]. Changes in brain plasticity as a result of disturbed neural redox homeostasis were detected in the brains of tumor-bearing mice with depressive-like behavior [46,47,48]. Structural evidence for a depressive-like state induced in a model of mammary cell carcinoma was also observed through decreased dendritic branching of pyramidal neurons in the medial prefrontal cortex [49].

Lipopolysaccharide is a component of gram-negative bacteria commonly used to induce a potent inflammatory response and behavioral changes that rapidly resolve within 24 h, followed by hyperalgesia [42]. Cytokine production in the tumor microenvironment can be detectable in the general circulation of experimental models of various tumor types, as well as in brain areas responsible for mood regulation. These studies reported increased plasma levels of IL-6, IL-12, TNF-α, IL-10, and IL-1β, but also increased IL-1β, IL-10 expression of IL-1β mitochondrial ribonucleic acid in the cortex and hippocampus, and increased levels of IL-6 and TNF-α in the hippocampus [50,51,52,53,54]. Hippocampal inflammation was related to depressive-like behavior in breast cancer mice, and also gastric-cancer-bearing mice with a significant increase in IL-6, IL-1β, reactive oxygen species (ROS), and cyclooxygenase-2 (COX2) [55,56]. The model of chronic stress and smoke exposure induced depression-like behavior and lung cancer, respectively, in mice, with the synergistic effect in a combined model manifested through a more prominent inflammatory response [57]. However, the impact of antidepressant fluoxetine was significantly attenuated under the conditions of chronic stress and LPS-induced inflammation, suggesting the role of chronic inflammation in the development of treatment-resistant depression [58]. We could identify several important underlying cascades in the development of depressiveness induced by inflammation.

Activation of inflammasomes, particularly nod-like receptor family pyrin domain containing 3 (NLRP3), may occur through DAMPs or PAMPs mediated by toll-like receptors (TLRs) and subsequently activate important intracellular pathways such as IFN I and the nuclear factor kappa-light-chain-enhancer of activated B cells (NF-κB) [34]. At the cellular level, repercussions of these processes could be the production of IL-1α, IL-1β, TNF-α, and IL-6, as well as the activation of microglia and the impairment of astrocytes in depression [34].

## 5. Underlying Inflammatory Disturbances in Carcinoma and Depression Co-Occurrence

Somatic illnesses could be followed by mental disturbances, or mental disorders could be a typical response to illness that vanished in reconvalescence with the illness resolution or could persist after somatic illness recovery [5,59]. Hart was the first to propose the concept that “sickness behavior” occurs as a short-term reaction in an acute inflammatory state and is crucial for the survival of the individual [60]. However, when inflammation becomes chronic, as in autoimmune diseases, neurodegenerative diseases, cardiovascular diseases, diabetes and obesity, and cancer, mood symptoms predominate and can worsen the disease.

Nearly 30% of cancer patients meet the criteria for a psychiatric diagnosis of depression, neurotic and stress-related disorders, adjustment disorders, sleep disorders, or delirium [61]. The problem of insomnia is very pronounced in patients in the active and stable phase of cancer, especially when associated with a pain syndrome and distress [62]. With regard to the onset and persistence of depressive symptomatology, it was very interesting for us to consider the overlap with pain and fatigue as symptoms of the cluster, presented as two or three concurrent and interrelated symptoms that may or may not have a common etiology and pathophysiological pathways [63,64]. Dodd et al. (2001) defined pain, fatigue, and insomnia in cancer patients as a cluster [65]. Recently, Charalambous et al. (2019) provided preliminary evidence that targeting fatigue, anxiety, and depression in patients with breast and prostate cancer may have a meaningful effect on pain as a related symptom [66]. A proposed underlying mechanism in the pathogenesis of these symptoms includes systemic inflammation with high pro-inflammatory cytokine levels, oxidative stress, and neuroendocrine-immune alterations [67,68,69,70,71,72]. Inflammation-mediated tryptophan catabolism along the kynurenine pathway might contribute significantly to the development of fatigue and depression in cancer patients [73].

Consideration of the common neuroimmune mechanisms of chronic pain and depression and the possible corrective anti-inflammatory effect of antidepressants seem to be of greater importance in this case [74]. Therefore, researchers have developed a model of inflammatory cytokine activity in cancer to explain the co-occurrence of pain, fatigue, and sleep disturbances [75] (summarized in Figure 1). Sometimes it is necessary to remember that the primary goal is to eliminate pain sensations to prevent the onset of depressive symptoms. Functioning could be especially compromised with pain sensations that are also correlated with ongoing inflammation [76]. Acute pain was also associated with acute inflammation, and chronic inflammation reflected chronic pain [59,77]. Chronic pain and depression in humans are associated with persistent low-grade inflammation rather than severe systemic inflammation, with only a partially common underlying mechanism [77]. Neuropathic pain has been shown to be associated with increases in the tryptophan-metabolizing enzyme indolamine 1,3 deoxygenase (IDO1) in the liver but not in the brain, and antagonism of the N-methyl-D-aspartate (NMDA) receptor by kynurenic acid [77]. On the contrary, co-morbid depression was mediated downstream of spinal cord IL-1β signaling and the formation of kynurenine and its metabolites in the brain [77,78].

Along with anxiety and depression, cancer-related fatigue is one of the most common symptoms in cancer patients [79]. Fatigue and depression have similar clinical presentations (Figure 1). Fatigue can occur independently, be a prodromal symptom of depressive disorders, or be part of a developed depression [80]. Fatigue is defined as a loss of energy that can affect physical, mental, or cognitive functioning and is manifested by loss of motivation, apathy, and reduced concentration and attention [81]. The above symptoms are important characteristics of depressive mood disorder. For these reasons, it is sometimes very difficult in clinical practice to distinguish whether it is just fatigue or depression. Recently, our research group has pointed out that acute and chronic inflammation have a significant impact on fatigue and depression in patients with the inflammatory and neurodegenerative disease multiple sclerosis. We observed that peripheral inflammation was related to fatigue and postulated that brain inflammation in acute episodes could further lead to neurodegeneration and mood and cognitive changes [70].

The new important clinical entity of paraneoplastic disorder should be considered in the context of the clinical field of autoimmune-mediated depression [82]. Paraneoplastic neurologic syndromes (PNSs) are rare cancer-related diseases that can affect any level of the central and peripheral nervous systems [83]. These disorders do not result from tissue invasion by the tumor, metastases, or metabolic or toxic effects of cancer therapy [84]. PNSs are caused by an immune response directed toward neural self-antigens aberrantly expressed by neoplastic cells and marked by specific autoantibodies [83,85]. Although PNSs can occur in any type of tumor, the most frequently associated malignancies include ovarian and breast cancer, small-cell lung cancer, thymoma, Hodgkin’s lymphoma, and neuroendocrine tumors [86]. The exact immunopathogenic mechanisms for most paraneoplastic syndromes are still unclear. The autoimmune theory postulates an immune cross-reaction between antigens expressed by tumor cells (“onconeural” antigens) and neurons [87]. The autoimmune response, initially directed against tumor cells, results in further damage to neurons that physiologically express the same antigen [86]. The target of the immune attack can be intracellular antigens (anti-Hu, anti-Yo, anti-Ma2, anti-Ri, GAD), antigens on synaptic receptors (NMDA, α-amino-3-hydroxy-5-methyl-4-isoxazolepropionic acid receptor, γ-aminobutyric acid receptor) or ion channels, and other cell-surface proteins (LGI 1, GQ1b) [88,89,90,91,92,93,94,95]. The main effector of the immune response in PNSs associated with antibodies directed against intracellular antigens is the CD8^+^ cytotoxic T cell, whose action results in rapid and extensive neuronal death by cytotoxic activity [96]. Mild signs of inflammation are commonly detected in the cerebrospinal fluid in the early phases of these disorders [97]. Antibodies against plasma membrane antigens, such as ion channels and surface receptors, may play a pathogenic role as direct effectors in neural tissue injury. Mechanisms by which these antibodies affect the targeted cells include antigen internalization and degradation, activation of complement cascades, antibody-dependent cell-mediated cytotoxicity, and blockade of receptor function [96]. Paraneoplastic syndromes of the CNS can be present with neuropsychiatric and cognitive symptoms, abnormal movements, new-onset epilepsy, and sleep disorders [98].

Over the past decade, evidence has accumulated of an intriguing relationship between cancer and neurodegenerative diseases. Progression of both conditions is primarily defined by a set of molecular determinants that are complementarily dysregulated or share important underlying biological mechanisms that promote cell proliferation and apoptosis, including alarmins (discussed in detail in [99]). DNA, cell cycle aberrations, redox imbalance, inflammation, and immunity are closely associated with shared characteristics of cancer and neurodegenerative diseases. The question arises whether each depressive episode and these kinds of repeated excessive immune and autonomic dysregulation could also contribute to neurodegeneration.

## 6. Potential for New Anti-Inflammatory Strategies in Cooccurrence of Carcinoma and Depression

The basic mechanism of action of conventional therapy for malignant diseases, such as radiotherapy and chemotherapy, is to induce the death of tumor cells [100]. However, the process of tumor cell necrosis is often triggered as an accompanying phenomenon in addition to the desired apoptosis. Necrosis is followed by the release of cellular contents outside the cell. Thus, endogenous alarmins reach the intercellular space and become inducers and facilitators of inflammation [100]. In this way, therapeutically induced tumor necrosis may be beneficial to the host [101,102]. Therefore, another no less important mechanism of action of the therapy is the induction of inflammation and the strengthening of the antitumor immune response. New therapeutically induced tumor necrosis may benefit the innate antitumor immune response, as necrotic cells facilitate the maturation of antigen-presenting cells [103,104]. Mature antigen-presenting cells, especially dendritic cells, induce a potent acquired antitumor response. Thus, the increase in systemic values of proinflammatory cytokines of innate immunity is accompanied by an increase in values of cytokines of acquired immunity. Chronic inflammation is present in and around most tumors, including those not causally related to an inflammatory process [105]. The percentage of patients with inflammatory components in the tumor microenvironment varied from 28% to 63% depending on tumor type [106]. Anti-tumor therapy is usually followed by a wave of acute inflammation that changes the intensity and course of the antitumor immune response [100].

Although radiotherapy and chemotherapy are options for the treatment of cancer, other treatments are increasingly being explored today, such as immunotherapy [107]. The use of monoclonal antibodies, immunomodulatory agents, modulated immunocompetent cells, or blocking antibodies for checkpoint molecules has shown significant results in cancer therapy and has fundamentally changed the approach to cancer therapy [108,109]. The discovery of checkpoint molecule inhibitors was awarded the Nobel Prize [110]. The blockade of cytotoxic T-lymphocyte-associated protein 4 (CTLA4) and programmed cell death protein 1 (PD1) molecules with antibodies is now very topical and has also found its application in clinical practice [111]. Research on blocking other checkpoint molecules such as T cell immunoglobulin and immunoreceptor tyrosine-based inhibitory motif domain (TIGIT), the cluster of differentiation 96 (CD96), natural killer receptor NKG2A is in full swing [112,113,114,115]. A strong effect of the application of this type of therapy is the enhancement of both innate and acquired antitumor immune responses [110,111,112,113,114,115]. This phenomenon is almost always accompanied by increased production of pro-inflammatory cytokines and momentum of inflammation in the host. These effects could be unwanted in the propagation of inflammation and consequently trigger depressive symptomatology.

Since alterations of various cytokines have been established in both depression and cancer, cytokine inhibitors deserve more detailed discussion. Infliximab, a TNF antagonist, improves depressive symptomatology by decreasing CRP levels [116] but has also shown beneficial effects in treating cancer-related fatigue [117]. Adalimumab, another TNF-α-specific neutralizing monoclonal antibody similar to infliximab, has been shown to significantly improve depressive symptomatology in patients with various chronic diseases [118,119,120], but without studies in psychiatric patients. Etanercept, another TNF-α antagonist, reduced depressive-like behavior in preclinical models, but also clinical studies in patients with psoriasis and rheumatoid arthritis [121,122,123,124,125]. Pentoxifylline, a methylxanthine drug that acts as a strong non-selective TNF-α inhibitor, has improved depressive behavior in animal models but has also shown positive results as an add-on treatment for depression [126,127,128]. Ustekinumab, an inhibitor of IL-12 and IL-23, dupilumab, an antagonist of the receptor of IL-4, ixekizumab, an IL-17A inhibitor, and guselkumab, an IL-23 inhibitor, have all been for their antidepressant action [129,130,131,132]. Although cytokine inhibitors have a more targeted effect on depression-related inflammation, these results were limited to specific patient groups. Because cytokine inhibitors are large molecules, they cannot cross the BBB, suggesting that their anti-inflammatory action is limited to peripheral TNF-α. This does not preclude their efficacy, but further studies are needed to determine their potential for treating depression in the presence of concomitant carcinoma.

Conversely, re-establishing balance in the peripheral secretion of cytokines is observed after antidepressant use and the resolution of depression. The most recent pharmacological protocols for the treatment of depression in carcinoma target monoamine neurotransmitters, brain-derived neurotrophic and inflammatory factors, and glutamate and its receptors, using monoamine oxidase inhibitors, tricyclic drugs, selective serotonin reuptake inhibitors (SSRIs) and selective serotonin noradrenaline reuptake inhibitors (SNRIs), glutamatergic drugs, opioids, and benzodiazepines [74]. In vitro, SSRIs have been shown to inhibit the release of TNF-α and NO from activated microglia, impede calcium ion influx, decrease the activation of the Janus kinase-signal transducer and activator of transcription (JAK-STAT) pathway, and also reduce inflammatory changes [133]. SSRIs and SNRIs decrease blood and tissue cytokines and regulate complex inflammatory pathways of NF-κB, inflammasomes, TLR4, and peroxisome proliferator-activated receptor gamma (PPAR-γ) [134]. Liu et al. (2020) showed in their systematic review and meta-analysis that patients with depression who responded to treatment had lower baseline levels of the chemotactic factor for neutrophils IL-8 than non-responders [135]. In addition, treatment with antidepressants decreases TNF-α and IL-5 levels. However, long-term treatment with SSRIs has been postulated to increase Th1 and decreases Th2-derived cytokines [136].

Celecoxib, nonsteroidal anti-inflammatory drugs, minocycline, but also statins, polyunsaturated fatty acids, pioglitazone, modafinil, corticosteroids, the vitamin D2 analog i.e., paricalcitol, etc. have already been reported as classical anti-inflammatory drugs with consequent antidepressant effects [137,138,139,140] (Table 1).

Celecoxib, a selective COX-2 inhibitor, exerts anti-depressive action by decreasing IL-6 expression and/or levels [137]. Minocycline, the second-generation tetracycline antibiotic, can cross the BBB more efficiently than other tetracycline antibiotics. It has anti-inflammatory, antioxidant, and neuroprotective effects within the CNS by preventing the release of inflammatory cytokines such as IL-6 and TNF-α [141]. It also inhibits neutrophil migration, degranulation, oxygen-free radical production, and NO release. Statins, known as lipid-lowering agents, have shown anti-inflammatory potency by decreasing levels of CRP and low-density lipoprotein (LDL) cholesterol, TNF-α and IFN-γ production in stimulated T cells, but also by reducing immune activation of T-helper cells [142]. Pioglitazone, primarily used as an antidiabetic drug, acts as a PPAR-γ agonist and decreases the expression of IL-1β, IL-6, TNF-α, inducible nitric oxide synthase (iNOS), and chemoattractant protein-1 (MCP-1/CCL2) [143]. It ameliorates depression-like behaviors by inducing the neuroprotective phenotype of microglia [144]. The psychostimulant modafinil reduces brain inflammation by impacting monocyte recruitment and activation, T cell recruitment and differentiation, cytokine production, and glial activation [145]. Corticosteroids, known for their anti-inflammatory properties, have also been studied for their antidepressant properties [146]. Because of their various side effects, which depend on their dosage and duration of treatment, they should be used with caution [147]. Paricalcitol, a vitamin D2 analog, regulates microglia-mediated neuroinflammation via decreased production of IL-1-β, inhibition of NF-κB and NLRP3 signaling, and caspase-1 overexpression [140]. In examining the link between depression and cancer, numerous experimental studies have revealed that activation of the kynurenine pathway of tryptophan degradation due to inflammation plays an important role in the evolution and persistence of both diseases [148].

The Hamilton group study showed that a history of depression, anxiety, and fear of tumor recurrence was associated with greater use of complementary treatment approaches [149,150]. The supplements most commonly used by patients are selenium (Se), folic acid, and omega-3 fatty acids [151]. Cancer patients often turn to antioxidants; among them, Se is particularly interesting, either from an inorganic source (sodium selenate) or the amino acid (selenomethionine) [151]. However, it is questionable whether its action can be considered exclusively as an antioxidant because it can also act as an oxidant and exhibit an anticarcinogenic effect [152]. Due to its antioxidant effect, Se is suitable as a supplement in depressive states and is an essential trace element for thyroxine metabolism. Thus, Se deficiency lowers antioxidant protection of the brain and may lead to brain damage—the turnover of dopamine and serotonin increases, while it decreases for noradrenaline and 5-hydroxy-3-indoleacetic acid compared to controls [153].

The role of folic acid is reflected in the synthesis of serotonin, and its supplementation is advised for patients with depression [154]. However, at high doses of folic acid, an adverse action may be observed because its role in metabolism controls the potential proliferative action for cancer cells [151].

Other widely used supplements are omega-3 fatty acids, predominantly eicosapentaenoic acid (EPA) and docosahexaenoic acid (DHA). They influence optimal cell structure/function and affect synaptic neurotransmission [155]. Therefore, they are recommended for complementary therapy in depression [156]. Improvements are also expected from fatty acid supplementation in chemotherapy and radiotherapy, as they affect inflammation, apoptosis, eicosanoid synthesis, etc. [157].

New therapeutic approaches may include drug-supporting/delivery systems as well as assorted supplements. Among the various supplements, zeolites are in the spotlight. There are a number of zeolite-associated positive effects reported in the literature: antioxidant and anticancer performance, ion exchange, and adsorption/encapsulation features, to name a few [158,159]. These aluminosilicates can be of synthetic or natural origin, such as clinoptilolite, and are recognized for human application [160]. Interestingly, synthetic zeolites can be designed to meet the specific demands of drug carrier systems and seem to be a far better choice, but are still awaiting general pharmaceutical recognition. Some therapeutic approaches may benefit from zeolite use, i.e., sustained drug-delivery systems, which are considered to be improved therapeutic pathways compared to regular ones [161]. Over the past two decades, researchers have competed to find ideal carriers, exploring a possible synergistic effect between the selected support and the drug itself [162,163]. There are several reasons for this—firstly, a specially designed carrier of nanometric dimensions must be considered to sustain BBB pass [164]. To meet this requirement, animal testing is set forward with some interesting applications. For example, infrared-activated BBB permeability may be accomplished by utilizing zeolitic imidazolate-based nanocomposite for intracerebral quercetin delivery providing neuroprotective effects [165]. Furthermore, the zeolite platform must encompass enough functional centers to efficiently adsorb/encapsulate drugs. Thus, zeolitic composites were proposed for synergetic tumor thermo-chemotherapy using doxorubicin drug delivery that sustains tumor reduction [166].

In the field of mental disorders, zeolite testing is under-explored with the majority of studies employing only animal models. One way to treat induced bipolar disorder in rats with probiotic cultures, alone and in zeolite-supported formulations, was suggested by Alchujyan et al. [167]. Interestingly, probiotics expressed a positive effect on arginase/nitric oxide synthase activities without significant benefits of zeolite carrier, as both formulations led to beneficial histopathological brain alterations and subsequent behavioral progress in rats.

Several reports suggest that recovery of cancer patients can be promoted by zeolite supplementation [168]. This hypothesis is based on zeolites’ excellent adsorption capacity for histamine which may be regarded as beneficial for pain relief [169]. In vitro and in vivo experiments on zeolite frameworks safety are extensively studied, while others investigated double-blinded trials of oral clinoptilolite intake in cancer patients to treat peripheral neuropathy induced by chemotherapy [170]. As reported by Vitale et al., the neuropathy extent was quite similar, occurring in 70.6% and 64.3% of patients in the placebo and zeolite supplementation groups, respectively [171].

Bearing in mind the good adsorption properties of zeolites, their role in the removal of heavy metals is often mentioned in the context of the prevention of mental disorders. A prospective use of zeolite/ethylenediaminetetraacetic acid as a lead scavenger is reported [172], confirming the role of clinoptilolite in reducing neurotoxicity in mice. Another removal of lead addresses issues with autism spectrum disorder [173]. Injection of zeolite particles is proposed with the possibility of stool excretion after metal adsorption, with no analysis of the detrimental effect zeolite nanoparticles could have on the hematological and gastrointestinal region. As a multifunctional material, Y zeolite is applied as an electrode support for the ruthenium ammine complex in the electrochemical detection of dopamine/serotonin [174]. Extending this system toward zeolite’s possible interaction with L-dopa, as a dopamine precursor may be sound due to several hydrogen bonds that can be formed. However, this emerged as a premise for rising dopamine levels, which is challenging to test/confirm [175]. Expectedly, these propositions are left with only hypothetical opinions.

## 7. Conclusions

Immune system alteration is the common denominator of depression and cancer. Additionally, alterations in the immune response seem to overlap in both pathological conditions. The psychiatric correlates are followed by immune disturbances, and we still wonder to what extent the resolution of inflammation in carcinoma might simultaneously contribute to the resolution of the associated depressive symptomatology. Recognition of acute mediators of inflammation is very important, and it is even more important to prevent the transition from acute to chronic inflammation through early anti-inflammatory interventions. The alarmins induce local (central) inflammation by TLR signaling and facilitating NF-κB transcriptional activity and NLRP3 inflammasome in neuronal and nonneuronal cells. Thus, pro-inflammatory cytokines produced in the periphery could activate inflammation in the brain and subsequently modulate the release and function of neurotransmitters, leading to the onset of depression. Previous clinical investigations have shown that the cytokines IL-1, IL-6, IFN-γ, and TNF-α play key roles in these processes. These same cytokines are among the major mediators of the anti-tumor immune response and the chronic inflammation that usually accompanies it. Hypersensitivity and chronification of inflammation suggest an exhausted and insufficient immune response. In conclusion, peripheral inflammation could trigger central immune-inflammatory pathways that lead to pain, fatigue, and depressive symptomatology in patients with cancer. Cancer treatment strategies, as well as conventional psychotropic drugs, could help balance the inflammatory milieu. The new equilibrium in both conditions may be achieved by variously targeted anti-inflammatory strategies. Anti-inflammatory drugs are well known, but new possible pathways and challenging add-on therapies have yet to be found.

## Figures and Tables

**Figure 1 cells-12-00710-f001:**
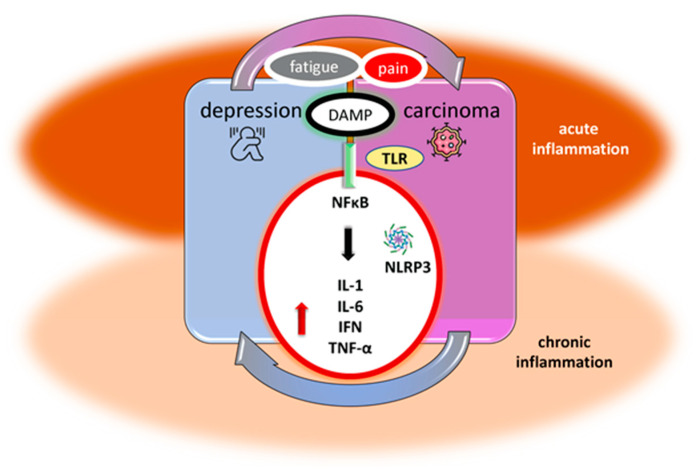
Inflammation as a substrate of depression and carcinoma co-occurrence. Peripheral inflammation is initiated with DAMPs through TLR, NLRP3, and NF-κB pathways. DAMPS are molecules released from damaged host cells that can be used as “alarmins”. Pro-inflammatory mediators such as acute-phase proteins, prostaglandins, leukotrienes, oxygen- and nitrogen-derived free radicals, chemokines, growth factors, and cytokines released by immune defense cells are particularly important for the early detection of acute inflammation. CRP, fibrinogen, and PCT as part of an innate immune response are detectable in serum within a few hours of inflammation initiation. The pro-inflammatory cytokines IL-1, IL-6, IFN, and TNF-α are elevated in carcinoma and consequently in the brain of patients with pain, fatigue, and depression. Acute and transitory inflammation must be distinguished from chronic and persistent inflammation. Anti-inflammatory strategies could lead to the resolution of inflammation and at the same time stabilization of depression and carcinoma. DAMP—danger-associated molecular patterns; TLR—toll-like receptors; NLRP3—nod-like receptor family pyrin domain containing 3; NF-κB—nuclear factor kappa-light-chain-enhancer of activated B cells; CRP—C-reactive protein; PCT—procalcitonin; IL—interleukin; TNF-α—tumor necrosis factor-alpha.

**Table 1 cells-12-00710-t001:** Drugs targeting inflammation with potential for joint antidepressant effects.

Drug	Class	Reference
Celecoxib	NSAIDs	[137]
Minocycline	tetracycline antibiotics	[138]
Statins	HMG-CoA reductase inhibitor	[139]
Pioglitazone	Antihyperglycemic	[139]
Modafinil	wakefulness promoting agents	[139]
Steroids	corticosteroids	[139]
Paricalcitol	vitamin D analog	[140]

Abbreviation: MAO—monoamine neurotransmitters; SSRIs—selective serotonin reuptake inhibitors; SNRIs—selective serotonin noradrenaline reuptake inhibitors; NSAIDs—non-steroidal anti-inflammatory drugs; HMG-CoA—β-hydroxy β-methylglutaryl-CoA.

## Data Availability

Not applicable.

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
