# Peer review of "Targeting Underlying Inflammation in Carcinoma Is Essential for the Resolution of Depressiveness"

_cells, 2023, doi:10.3390/cells12050710_

Round 1
Reviewer 1 Report
Comments in the attachment. This work should be thoroughly edited. Try to make it more complex, go one step forward.
Review Report: cells-2152580
TARGETING UNDERLYING INFLAMMATION IN CARCINOMA IS ESSENTIAL FOR THE RESOLUTION OF DEPRESSIVENESS
Borovcanin et al. explain in this review the similarities in inflammation between oncological diseases focused on carcinoma and depressiveness, and the interplay between both conditions. They propose anti-inflammatory drugs as a tool to help to resolve cancer-associated depressiveness.
Notes:
1. Introduction:
- More literature needed, mainly after reference [5], where it is explained what it is stated afterwards, i.e., the hallmarks of both oncological and mental disorders, and where it must be demonstrated that exacerbation of somatic and mental disorders are related to acute or chronic inflammation.
2. Acute and chronic inflammation in carcinoma
- References are too disorganized, repeated, and chaotic. What I suggest is to reorganize the information so the literature could be better organized, and add more various literature.
- Last paragraph: improve English and make the language more formal.
3. Acute and chronic inflammation in depression
- Line 114: change reference citation to Majd et al. (2020) so it will have a more consistent format. Moreover, change the year of publication to the correct date 2020.
- Line 120: improve general redaction of the sentence for more clarity. Perhaps the intention was saying that ‘they based their conclusions on Capuron et al. (2002) who demonstrated that IFN administration […] causes neurovegetative symptoms at first […]’.
- Line 137: remove comma.
4. Animal models of inflammatory-induced depression in carcinoma
- Try to add more references instead of repeating too many references cited before. Keep going on the exposition of new data from other researchers or other publications instead of only present few references.
5. Underlying inflammatory disturbances in carcinoma and depression co-occurrence.
- Figure 1: glossary of abbreviations should be more consistent in terms of using the same format, e.g., TNF-α should appear before the complete name of the factor, not in brackets afterwards.
6. Potential for new anti-inflammatory strategies in cooccurrence of carcinoma and depression
- Be more precise when talking about “chronic inflammation is almost always present”. Provide a percentage and reference that supports the information.
- Add missing abbreviations for gaining consistence, e.g., CTLA4, PD1, PPAR-γ, etc.
- “Liu et al.” format for consistence.
- Table 1 title: be more explicative of what you are showing. Celecoxib is actually an anti-inflammatory drug, so the “potential anti-inflammatory mechanism of action” is quite obvious. Be clearer about you want to show with this table.
7. Conclusions
- Bad numbering of section title: it must be 7.
- Good explanation of the objective of the review. I suggest including it in the introduction as well, and be clearer with that aim in the abstract.
- The statement “we are wondering if resolving the inflammation could resolve the depression” may be too pretentious in the absence of preliminary data as depression is a complex mental disorder and its molecular basis are mostly unknown. Suggestion: change to “if resolving inflammation in carcinoma could help to resolve some associated depressive disorders”.
Author Response
Please see the attachment:

Reviewer 2 Report
Manuscript "Targeting underlying inflammation in carcinoma is essential for the resolution of depressiveness" is interesting and well done written. It includes principles of inflammation, depression and carcinoma.
It is divided in appropriate sections and target importance of each of them.
Understanding and connection between inflammation, depression and carcinoma is old-new issue. Till now, we do not know what is older inflammation or carcinoma. We have a lot of controversies about this topic. Also depression could be result of previous processes and it is tightly connected with previous processes. This manuscript tried to summarize all works in these topics with novel references, novel view of reflection and good knowledge and background of the previous knowledge.
I have few minor corrections and questions for authors.
1. Can you rewrite manuscript to be easier for reading? There is a lot of overlapping and it is not easy to understand the point of all these important interesting topics.
2. There is lot of confused and thrown information about inflammation, depression, fatique. Can you divide them?
3. Paragraph “Activation of inflammasomes, particularly nod-like receptor family pyrin domain containing 3 (NLRP3), may occur through DAMPs or PAMPs mediated by toll-like receptors (TLRs) and subsequently activate important intracellular pathways such as IFN I and nuclear factor kappa-light-chain-enhancer of activated B cells (NF-κB), leading to cell production of IL-1α, IL-1β, TNF-α, and IL-6, as well as activation of microglia and impairment of astrocytes in depression.” This is confused and schematized. It should be changed.
4. Acute pain was also associated with acute inflammation, and chronic inflammation reflected chronic pain [41,48]. Pain threshold was associated with kynurenine signalling pathway and IL-1β secretion in the liver and peripheral and central nervous systems, suggesting similar underlying mechanisms of allodynia and depression. – describe in a more propriate way with other pathways and mechanisms which are included in this process.
5. Figure 1. Inflammation as a substrate of depression and carcinoma cooccurrence.
Explanation of the good scheme is confusing. I suggest to explain more about earlier parameters of inflammation especially mediators in acute inflammation. It would be better to emphasize more biomarkers of acute inflammation in order to find it as predictors in earlier phases of these disorders.
6. Section “Acute and chronic inflammation in carcinoma” – explain and point on the acute inflammation. Parameters of oxidative stress and inflammation are usually detected in the chronic phase of diseases. Is there any parameters which can be detected earlier?
7. Section “Potential for new anti-inflammatory strategies in cooccurrence of carcinoma and depression”
Describe more about anti-inflammatory mechanisms of mentioned drugs. Report more groups of drugs with antiinflamatory action. Point their use in the context of the title of manuscript.
8. Mention more supplements beside zeolite with antioxidant and anticancer performance.
9. Conclusion is confused and does not have the point of the conclusion. Please, rewrite it.
10. There are mistakes in reference list. For example ref number 52.
Author Response
Please see the attachment:

Round 2
Reviewer 1 Report
The manuscript has been broadly improved according reviewers' reports and some extra improvements were also done. Good rephrasing, fine corrections, great new data, good organization, clear objective and approaching.
Also appreciated the coverletter from the authors.
CORRECTIONS:
- Text below Table 1: According to provided literature (137) and extra literature I checked myself, celecoxib DECREASES IL-6 levels.
- Make sure that the numbering of the conclusion epigraph is correct because it is not seen properly in the manuscript.
Author Response
Please see the attachment:

Reviewer 2 Report
The authors give a rationale for this work and discussed the results in the light of recent findings. Your manuscript is highly improved and satisfied.
I suggest to romove the first sentence from your conclusion.
Author Response
Please see the attachment:
